# The Influence of Red Light (RL) and Effective Microorganism (EM) Application on Soil Properties, Yield, and Quality in Wheat Cultivation

**Mariusz Szymanek** [1,*], **Agata Dziwulska-Hunek** [2], **Janusz Zarajczyk** [3], **Sławomir Michałek** [4] **and Wojciech Tanaś** [1]

1   Department of Machine Science, University of Life Sciences in Lublin, 20-612 Lublin, Poland; wojciech.tanas@up.lublin.pl
2   Department of Biophysics, University of Life Sciences in Lublin, 20-950 Lublin, Poland; agata.dziwulska-hunek@up.lublin.pl
3   Department of Horticulture Machinery and Tools, University of Life Sciences in Lublin, 20-950 Lublin, Poland; janusz.zarajczyk@up.lublin.pl
4   Department of Botany and Plant Physiology, University of Life Sciences in Lublin, 20-950 Lublin, Poland; slawomir.michalek@up.lublin.pl
*   Correspondence: mariusz.szymanek@up.lublin.pl; Tel.: +48-81-531-97-37

**Abstract:** The aim of the study was to determine the impact of red light (RL) and effective microorganisms (EMs) on the wheat yield of grain and straw, as well as the quality (protein, carbohydrates, gluten, index of sedimentation (SDS index), germination capacity). Moreover, the experiments measured the granulometric composition and pH of soil, as well as its content of microelements and major nutrients, including heavy metals. The field experiment was conducted in 2017, 2018, and 2019 at the experimental station (Felin) of the University of Life Sciences in Lublin (Poland). The best results in terms of overall yield, protein content, and SDS index were obtained after the red light treatment (RL). The three-year application of effective microorganisms (EMs) in the soil had a positive impact, relative to the control, on the grain yield, straw yield, starch, SDS index, and germination capacity. A slight decrease was observed in terms of the protein content. After the application of effective microorganisms (EMs) in soil, changes were observed in the soil's granulometric composition, pH, humus, and microelements.

**Keywords:** halogen lamps; beneficial bacteria; germination rate; chemical composition of grain; granulometric composition of soil

## 1. Introduction

The contemporary development of agriculture is strongly dependent on the development of new agrotechnical treatments and the rational use of agricultural space, with a particular focus on skillful and safe increases in the yields generated from crops [1].

However, the drive towards increased productivity requires the involvement of extensive production resources and may have a negative impact on the environment [2].

The search for the optimum methods of increasing production capacity per a unit of land area, coupled with the increasing limitations imposed on certain industrial agricultural solutions (mineral fertilizers, pesticides, etc.), turns a lot of eyes towards the potential offered by effective ecological agriculture [3].

One of the key factors determining the development, growth, and yield of plants is the quality of seeds. The pre-sowing treatment of seeds aims to improve their germination capacity, as well as the vigor of seedlings. Young plants with greater vigor develop faster and are more resilient to difficult

(stressful) environmental conditions. Hence, crop seed production is generally focused on obtaining better sowing material through seed refinement. To date, the standard methods of processing sowing material in agricultural practice have relied primarily on the use of various seed treatment preparations, chemical fertilizers, pesticides, etc., which more often than not include toxic ingredients [4–9].

In the late 1980s, the first attempts were made to also employ physical stimuli for such purposes, including: laser light, magnetic fields, electrical fields, etc. Yet, despite numerous studies conducted and published in this context [10–19], the exact mechanism responsible for the impact of physical factors on plant material remains undiscovered.

One of the seed biostimulation methods involves radiation using light of a proper wavelength. It is possible due to the existence of a specific, complex energetic system in plants which absorbs, transforms, stores, and utilizes the energy of photons falling onto the plant's organs [20]. Such stimulation leads to earlier maturation and elevated yields of plants, and to the occurrence of an increase in stored sugar and protein levels [21]. The research results obtained so far indicate a beneficial effect of the pre-sowing red light stimulation of seeds [22–26].

Another possible method of refining seeds entails treatment with the use of EM biopreparations (effective microorganisms, EM farming, ProBio EM) containing a mixture of beneficial microorganisms. EMs are a composition of 80 aerobic and anaerobic microorganisms (milk fermentation and phototrophic bacteria, actinomycetes, yeast, and fungi) [3,27], carefully selected representatives of the world's smallest organisms. EM technology was first introduced by Teruo Higa, a professor of horticulture at the University of the Ryukus in Okinawa, Japan [27–32].

Effective microorganisms include photosynthetic bacteria (*Rhodopseudomonas* spp.), lactic acid bacteria (*Lactobacillus* spp.), and yeast (*Saccharomyces* spp.). Photosynthetic bacteria absorb sunlight and soil heat energy and subsequently convert root system exudates, organic soil fraction, and gases (ammonia) into cell-building compounds (amino acids, nucleic acids, and sugars). The products may be absorbed directly into the plants, which affects their growth and the soil ecosystem. Moreover, they coexist with other bacteria—*Azotobacter* and *Rhizobium*—capable of binding atmospheric nitrogen [26]. Lactic acid bacteria in EMs are capable of producing lactic acid from sugars and carbohydrates provided by photosynthetic bacteria and yeast. The bacteria have sterilizing properties and protect plants against disease [33,34]. Yeast serves the role of producing hormones and enzymes. It could be said that the effects of all the bacteria and yeast are, in a way, complementary [35].

An EM biopreparation is a complex provided with specialized biological tools (enzymes) to facilitate the survival and development of the microorganisms, not only in soil, but also in a variety of other environments (landfills, bottom sediments in water reservoirs, bedding in livestock buildings, manure pits, sewage drains, and sewage sludge at water treatment facilities) [32]. The use of EMs has become commonplace in various aspects of our everyday lives: horticulture, environmental protection, medicine, and many other industry sectors [18]. Until quite recently, it was believed that the coexistence of aerobic and anaerobic organisms that normally thrive in extremely different environments could not be accomplished [17]. Aerobic microorganisms help to sustain and develop life, whereas anaerobic organisms decompose harmful or already processed substances [36].

The mechanism of some microorganisms' activity entails the conversion, modification, and utilization of toxic impurities to generate energy and produce biomass. The process is sometimes referred to as bioremediation or biodegradation [37]. Microorganisms can be a valuable repair tool that aids the elimination of impurities from soil, water, and sediments [38].

Bioremediation is the process of eliminating and modifying toxins present in chemical substances and physical waste from the environment by employing the activity of bacteria, fungi, and plants. Moreover, microorganisms have the ability to spread evenly in an environment. The activity and growth of microorganisms is affected by the pH, temperature, moisture, and structure of soil and solubility in water, as well as the availability of nutrients and oxygen. They offer an environmentally friendly solution with promising potential in countering a variety of environmental threats [39].

Scientific studies focusing on the impact of an EM biopreparation's application in the soil environment, as well as on the growth, development, and yield of plants [2,5,27,30,40–42], and in terms of general yield improvement specifically in the Polish context, have been few and far between. A vast majority of publications rely on information cited from research conducted under conditions (climate, soil, plants, soil organisms) distinctly different from those existing in Poland. In turn, the observations of farmers who typically have little to no affinity for scientific research can lack the necessary objectivity [43]. The use of physical methods and EM biopreparations has a positive impact on both the environment and the growth, development, and yield of crops.

Nonetheless, the literature is still lacking sufficient information pertaining to agrotechnical seed conditioning with the use of light and EM preparations to allow for the full understanding of the processes involved and, e.g., the specification of the most adequate dosages. Most of the available EM studies were conducted in the short term, on a small sample of plants, or in combination with mineral fertilizers. Despite the continuous development of our scientific and technical knowledge, many processes taking place in plants remain a mystery to us. This fact inspired us to consider the research problem discussed herein.

The aim of the study was to determine the impact of pre-sowing seed conditioning with the use of red light (RL) and an effective microorganisms (EM) biopreparation on the germination rate, grain and straw yield, protein, starch, gluten, and index of sedimentation (SDS index) of wheat grain, and the impact of EM application on soil properties (granulometric composition, humus content, pH and microelements).

## 2. Materials and Methods

### 2.1. Plant Material

An experiment was conducted on wheat of three cultivars (Bogatka, Tonacja, Zyta). The cultivars can be grown anywhere in Poland, including on weaker soils. They show medium tolerance to soil acidification [44]. The field experiment was conducted in 2017, 2018, and 2019 at the Experimental Farm of the University of Life Sciences in Lublin, located in Felin (Lublin, Poland) (51°13′21.9″ N, 22°37′55.85″ E). The soil at the Experimental Farm is composed of light salty clay loam and classified as good wheat complex. The grain yield was collected from an experimental area of 30 m$^2$.

### 2.2. The Method of Conditioning Seeds with RL and the EM Biopreparation

Before sowing, the seeds were treated with: RL, EMs, and a combination of the two methods, first conditioned with red light and subsequently with the EM biopreparation (RLxEM). Seeds not conditioned with either RL or the EMs biopreparation constituted the control sample.

One day before sowing, the grain was treated with light generated by halogen floodlights at a wavelength of 650–670 nm and with a stream density of 110–130 W·m$^{-1}$. The exposure time for a single wheat grain was 0.1 s.

The method of the pre-sowing seed conditioning entailed exposing both sides of a single layer of grains directly to RL.

The device used for this purpose has been presented and described in [22].

The EM biopreparation was applied annually into the soil at a dose of 40 l·ha$^{-1}$

### 2.3. Granulometric Soil Composition

The seeds were sown into soil without EM biopreparation treatment, control (1), or with EMs content, (2). Each research object was harvested and evaluated in triplicate. Soil characteristics in terms of its granulometric composition were determined with the pycnometric method in accordance with PN-R-04032 [45] (mean after three years). After the mineralization of the soil, the following methods were used to conduct the appropriate chemical analyses: pH—potentiometrically in distilled water; microelements, humus—Tiurin's method; bioavailable forms (KCL, P2O5, K2O, Mg), bioavailable

microelement forms (B, Cu, Mn, Zn, Fe); heavy metals (Cd, Pb, Ni, Zn, Cu, Mn, Fe, Cr, As, Hg)—atomic absorption spectroscopy with flame and electrothermal atomization.

## 2.4. Yield and Grain Quality

Plants were cultivated in accordance with the recommended techniques. The analysis pertained primarily to the yield and quality of grain (germination capacity, content of some nutrients). The germination capacity was assessed at the Instytut Hodowli Nasion i Aklimatyzacji Roślin (Seed Production and Plant Acclimatization Institute) in Radzików. The seeds were sown onto Petri dishes covered with blotting paper moistened using distilled water [46].

## 2.5. Chemical Composition of Wheat Grains

The analysis of chemical composition was performed with the Kjeldahl method [47], in terms of the protein content, in accordance with PN-EN ISO 11215:2002 [48] for starch, PN-A-74042/02:1993 [49] for gluten, and PN-EN ISO 5529:2010 [50] for the sedimentation index, to determine the quality of proteins.

## 2.6. Statistical Analysis

The obtained results were evaluated with the use of variance analysis. Where significant differences were confirmed on the basis of the significance test F statistic, quantitative inference was performed relative to Tukey's confidence intervals for the significance level of $p < 0.05$. The accuracy of respective measurement results was determined by providing the additional 95% confidence intervals for the arithmetic mean. The calculations were conducted with the use of Statistica 12.0 PL software.

## 3. Results

### 3.1. Impact of the EMs Biopreparation on Soil Properties

The analysis of the soil's granulometric composition (Table 1) indicated that it changed over time, meaning that the process of pedogenesis was ongoing.

**Table 1.** Granulometric composition of soil fraction content (%) (mean from three years).

| Objects | Humus Content (%) | Sand | Silt | Clay |
|---|---|---|---|---|
| | | 2.00–0.05 mm | 0.05–0.002 mm | <0.002 mm |
| 1 | 1.94a | 26.0b | 66a | 8a |
| 2 | 1.08b | 30.5a | 63b | 6.5b |

1—control field with no EM in the soil; 2—field treated with EM. Mean values in a given column designated with the same letter are not significantly different from each other according to Tukey's multiple range test.

Silt was the dominant fraction. Three years after the beginning of the experiment, it was observed that the percentage of the sand fraction was on the increase, while those of silt and clay decreased. The obtained results indicate that the use of the EM biopreparation had a statistically significant impact on the granulometric composition of the soil.

The analyzed soil's reaction was acidic (Table 2). After three years of EM application, the acidity of the soil, as well as the content of magnesium, phosphorus, and potassium, significantly changed.

**Table 2.** Reaction of the soil and content of bioavailable forms (mg·100 g$^{-1}$ of soil) (mean from three years).

| Objects | pH in 1 mL KCL | P$_2$O$_5$ | K$_2$O | Mg |
|---|---|---|---|---|
| 1 | 4.97a | 11.00a | 16.70a | 4.55a |
| 2 | 4.62a | 10.90a | 12.35a | 3.90a |

1—control field with no EM in the soil; 2—field treated with EM. Mean values in a given column designated with the same letter are not significantly different from each other according to Tukey's multiple range test.

The microorganisms reduced the concentration of phosphorus in 100 g of soil by 0.1 mg, which corresponded to a 1.0% change. Similar to phosphorus, the content of potassium decreased after EM application. Relative to the control, the value decreased by 4.35 mg (which corresponded to 26%) after 3 years. The content of magnesium was reduced to 3.90 mg/100 g of soil, thus indicating a decrease by 0.65 mg, i.e., 14%.

The content of the element manganese showed clear fluctuations relative to the control and after three years EM application (Table 3). A 22% increase in the soil manganese content after three years was observed. In the case of the other tested elements, i.e., iron and copper, an increase of, respectively, 3.6% and 5.2% was recorded.

**Table 3.** Content of bioavailable microelement forms (mg·kg$^{-1}$ soil) (mean of three years).

| Objects | B | Cu | Mn | Zn | Fe |
|---|---|---|---|---|---|
| 1 | 0.51a | 1.53b | 111.3b | 6.16a | 740.0b |
| 2 | 0.50b | 1.61a | 139.0a | 5.83b | 766.5a |

1—control field with no EMs in the soil; 2—field treated with EMs. Mean values in a given column designated with the same letter are significantly different from each other according to Tukey's multiple range test.

The content of heavy metals in the soil was significantly affected both by the duration of the experiment and the EMs conditioning (Table 4).

**Table 4.** Content of heavy metals (mg·kg$^{-1}$ soil) (mean of three years).

| Objects | Cd | Pb | Ni | Zn | Cu | Mn | Fe | Cr | As | Hg |
|---|---|---|---|---|---|---|---|---|---|---|
| 1 | 2.7a | 10.66a | 9.36a | 26.44a | 5.09a | 368b | 8725a | 12.52a | 1.41b | 0.038a |
| 2 | 2.7a | 10.08b | 8.24b | 25.36b | 5.03b | 583a | 8150b | 12.79b | 1.59a | 0.042b |

1—control field with no EMs in the soil; 2—field treated with EMs. Mean values in a given column designated with the same letter are not significantly different from each other according to Tukey's multiple range test.

The experimental factors had no significant impact on the content of cadmium in the soil. As for the other elements, both decreases and increases in the respective content values in the EM-treated soil were observed. Increases were recorded for: manganese, chromium, arsenic, and mercury, with the highest value of 58% observed for manganese. It is noteworthy that the content of heavy metals in the soil, including nickel, zinc, and copper, decreased by between 1.2% and 12% under the influence of effective microorganisms.

*3.2. Wheat Yield and Its Quality*

Table 5 presents the results obtained after the application of RL, EMs, and both treatments simultaneously on the yield of straw and grain, as well as the quality thereof.

**Table 5.** The influence of red light (RL) and effective microorganisms (EMs) on grain yield, straw yield, protein, starch, gluten, index of sedimentation (SDS index) and germination (mean of three years).

| Objects | Grain Yield, kg·m$^{-2}$ | Straw Yield, kg·m$^{-2}$ | Protein, % | Starch, % | Gluten, % | SDS Index | Germination, % |
|---|---|---|---|---|---|---|---|
| | Bogatka | | | | | | |
| 1 | 0.954d | 1.038d | 12.15c | 59.70b | 28.68d | 27.50d | 94.0d |
| | (0.004) * | (0.004) | (0.040) | (0.040) | (0.025) | (0.042) | (0.053) |
| 2 | 1.968a | 1.088c | 12.64a | 59.49c | 29.68b | 31.00a | 96.0c |
| | (0.004) | (0.003) | (0.026) | (0.028) | (0.026) | (0.078) | (0.068) |
| 3 | 1.046c | 1.183b | 11.99c | 59.84a | 30.20a | 29.00c | 97.5b |
| | (0.006) | (0.002) | (0.120) | (0.026) | (0.064) | (0.101) | (0.058) |
| 4 | 1.061b | 1.286a | 12.35b | 59.75b | 28.89c | 30.50b | 98.5a |
| | (0.002) | (0.007) | (0.051) | (0.027) | (0.025) | (0.251) | (0.041) |
| *p*-value | 0.000 | 0.000 | 0.000 | 0.000 | 0.000 | 0.000 | 0.000 |
| | Tonacja | | | | | | |
| 1 | 0.915a | 1.064d | 12.16c | 59.68a | 28.09d | 29d | 95d |
| | (0.004) | (0.003) | (0.031) | (0.036) | (0.024) | (0.065) | (0.051) |
| 2 | 1.097c | 1.397b | 12.10a | 45.95b | 28.64b | 30c | 98a |
| | (0.054) | (0.002) | (0.021) | (0.028) | (0.054) | (0.084) | (0.049) |
| 3 | 1.174a | 1.492a | 12.30b | 59.61a | 29.10a | 30.5b | 97.5b |
| | (0.004) | (0.005) | (0.042) | (0.034) | (0.037) | (0.34) | (0.041) |
| 4 | 1.227b | 1.372c | 12.33a | 59.35c | 28.54c | 31.5a | 97c |
| | (0.005) | (0.004) | (0.039) | (0.023) | (0.031) | (0.47) | (0.037) |
| *p*-value | 0.000 | 0.000 | 0.000 | 0.000 | 0.000 | 0.000 | 0.000 |
| | Zyta | | | | | | |
| 1 | 0.933c | 1.453a | 12.66a | 45.72d | 30.72d | 37.5c | 97.0c |
| | (0.004) | (0.007) | (0.029) | (0.026) | (0.034) | (0.054) | (0.034) |
| 2 | 0.885d | 1.218d | 12.90b | 46.02c | 31.21b | 37d | 97.5b |
| | (0.004) | (0.003) | (0.036) | (0.037) | (0.031) | (0.024) | (0.032) |
| 3 | 0.994b | 1.348c | 12.96a | 58.81b | 31.04c | 38b | 96.5d |
| | (0.004) | (0.005) | (0.029) | (0.024) | (0.028) | (0.035) | (0.051) |
| 4 | 1.093a | 1.427b | 12.77c | 58.94a | 31.85a | 38.5a | 99a |
| | (0.004) | (0.004) | (0.031) | (0.037) | (0.026) | (0.012) | (0.037) |
| *p*-value | 0.000 | 0.000 | 0.000 | 0.000 | 0.000 | 0.000 | 0.000 |

* Standard deviation; 1, control—no application of effective microorganisms (EMs) and irradiation by red light (RL); 2—irradiation by RL; 3—effective microorganisms (EMs); 4—RLxEMs. Mean values in a given column designated with the same letter are significantly different from each other according to Tukey's multiple range test.

The research showed that both RL and EMs, as well as RLxEms, increased the grain yield in the case of the Bogata, Tonacja, and Zyta varieties. However, for Zyta, an increase was observed for EMs and R x EMs, and a decrease for RL.

With regard to the control, the highest increase in grain yield (ca. 106%) was obtained for the Bogatka variety when using RL. In the case of the Zyta variety, using RL resulted in a reduction of grain yield by ca. 5%. The highest increase (ca. 28%) was obtained for the Tonacja variety when applying EMs. In the case of the Zyta variety, the application of EMs resulted in a reduction of the straw yield by ca. 7%. Regarding protein content, the highest increase (ca. 4%) was observed for the variety Bogatka for the RL treatment, and the lowest (about 1%) for the Zyta variety with RLxEms. In relation to the Bogatka variety, the EMs application reduced the protein content by about 1.3%, and for the Tonacja variety, the use of RL reduced the protein content by about 0.5%. An increase in the starch content was observed in relation to the EM and RLxEMs application groups by about 0.23% and 0.1%, respectively, for the Bogatka variety, and for the RL, EM and RLxEMs application groups by about 0.65%, 28.6%, and 29%, respectively, for the Zyta variety. However, a reduction of starch content by about 0.35% occurred under the influence of RL for the Bogatka variety. In the case of the Tonacja variety, the use of RL, the application of EMs, and the combination of RLxEMs resulted in a reduction of starch from

about 0.1% (EMs) to about 23% (RL). The obtained results showed that the RL, EMs, and RLxEMs treatments increased the gluten content. The highest growth (ca. 3.6%) was observed for the Zyta variety, and the lowest (ca. 0.73%) for the Bogotka variety with RLxEMs.

The highest increase (ca. 12.7%) in relation to the SDS index was obtained for Bogatka, and the lowest (ca. 1.3%) for Zyta with RL. On the other hand, with regard to germination, the highest increase (ca. 4.8%) was noted for the cultivar Bogatka with RlxEMs, and the lowest (ca. 0.5%) for the cultivar Zyta with RL.

## 4. Discussion

One of the main properties of soil is its texture. Heavy soils are characterized by high colloid contents. This determines the monolithic development of the structure, which is not welcome from the perspective of agricultural practice. Moreover, under low humidity conditions, such soils tend to dry up quickly, which can cause damage to plant roots. In studies by Tołoczko et al. [51] and Gajewskiego [52], the use of EMs preparations had no significant impact on the structural properties of soils with light and medium consistencies.

As follows from literature reports, the physical and chemical properties of soil are also improved after the introduction of probiotic bacteria, with the actual results depending on the type of soil and the dosage of the preparation [3,53]. Kucharski and Jastrzębska [54] observed that the use of effective microorganisms inhibited the growth and development of fungi and other soil bacteria in wheat cultivations. In turn, Badura [55] concluded that the use of microorganic preparations would be effective in degraded soils.

Radkowski and Radkowska [3] observed in the course of a three-year study that the use of an effective microorganism preparation changed the phosphorus and potassium content in the soil, but had no significant impact on the pH or magnesium content. EMs fertilization also improved the content of major nutrients (with the exception of sodium) and facilitated the intake of copper, zinc, manganese, and iron in multiannual plants (grasses, red and white clover). The authors also noted that, after the first year, an increase in boron, copper, manganese, zinc, and iron was observed, while after three years, only the manganese levels remained elevated. Conditioning soil with EMs resulted in an approx. 40% increase in manganese content.

A positive impact of the EMs preparation, leading to a 23% increase in wheat grain yield, was observed by Piskier [56]. In our study, the employed treatment regimens entailing the use of effective microorganisms (EMs) and a combination of red light and effective microorganisms (RLxEMs) increased the yields of both grain (9.6–11.2%) and straw (14–24%).

In a study by Gawęda et al. [57], the use of a double dose of EMs resulted in a 24% increase in soya yield relative to the control.

A positive impact of EMs treatment in combination with various fertilization regimes on the growth, yield, and quality of corn was observed by Shah et al. [40]. The yield of corn kernels increased from 69% (37.50 kg P + 60 l EMs ha$^{-1}$) to 284% (150 kg N + 75 kg P + 30 l EMs ha$^{-1}$). Simultaneously, the protein content in the kernels changed by between 1% (75 kg N + 60 l EMs ha$^{-1}$) and 9% (75 kg N+ 37.50 P + 60 l EMs ha$^{-1}$) [40].

It is noteworthy that conditioning soil with effective microorganisms does not always correlate with better plant growth, yields, or resistance to pests [58,59].

The application of an EMs preparation to the soil resulted in a 50% decrease in the yield of mung bean, while the best effects were obtained when it was combined with manure or the recommended dosage of mineral fertilizer [28]. Shaheen et al. [60] observed a 5.5-fold increase in spinach yields after the use of effective microorganisms combined with various fertilizers. In a study conducted under greenhouse conditions by Megali et al. [59], the use of EMs in various plant species produced significantly increased biomass yields in the case of alfalfa, corn, lettuce, pea, spinach, and tomato plants while, at the same time, no significant changes were observed for carrot, onion, or wheat.

Various physical, environmentally friendly methods are used with the view of improving sowing material. The impact of physical factors on plants depends on the type of radiation employed,

as well as various characteristics of the plants themselves, including species, cultivar, age, etc. [61]. The most common methods entail the use of a laser of halogen light. Still, despite a considerable body of conducted research, the actual mechanism of the light's influence remains unclear. The plant photochrome absorbs light at wavelengths of approx. 650–670 nm (red range). The generated energy is stored and later converted into chemical energy, such as heat, which facilitates the further development of the plant [62].

Abu-Elsaoud et al. [63] conducted a study on the effect of laser light (infrared) with various exposure times on grains of wheat. The highest germination capacity was observed for 1200 s exposure, where the increase reached 56% relative to the control. In our study, the wheat germination rate relative to the factors used (RL, EMs, and RLxEMs) increased by 2.1%, 3.7%, and 4.8% for Bogatka and by 3.2%, 1.6%, and 2.1% for Tonacja, respectively. However, for Zyta, an increase of 0.5% with RL and 2.1% with RLxEMs and a decrease of 0.5% with EM were observed.

In a study by Rassam et al. [64], where dry and moist wheat seeds were treated with various types of laser light (incl. He-Ne, diode) under different exposure times (1, 5, 10, and 15 min), the germination rate of wheat increased by 19% and 21% (respectively: moist seeds, 5 min exposure and dry seeds, 1 min exposure) for the He-Me laser, and by 16% and 19% (respectively: moist seeds, 5 min exposure and dry seeds, 1 min exposure) for the diode light. Changes to parameters such as the type of the laser, the wavelength of light, and the time of exposure were found to be capable of both stimulating and inhibiting the germination of wheat seeds. The use of laser light with a 980 nm wavelength and exposure times of 30 and 60 s had a positive impact on germination and reduced the incidence of incorrectly germinating grains (e.g., seed leaves emerging instead of roots or roots not growing and rotting away) [65].

Other physical methods have also been used in this context, e.g., ionizing radiation treatment of wheat seeds. Studies revealed that gamma radiation inhibited the growth of wheat, as well as reduced the germination and sprouting rate [66]. In turn, El-Kameesy et al. [67], in a study on the impact of gamma radiation (treatment dose—krad, from 0 to 10) on wheat germination (in the period from 1 to 7 days), observed an change (relative to the control, percentage of germination 20%) from 10 (10 krad) to 25% (2.5 krad) after 1 day and from 40 (10 krad) to 100 (control, 3.0 and 5.0 krad). Qiu et al. [68] reported increased protein content in wheat leaves after exposing sowing material to He-Ne laser light treatment. The quality of wheat grain depends primarily on the protein content and gluten durability. Gluten is composed of two proteins: glutenin and gliadin. The increase in grain protein content is significantly affected by nitrogen fertilization [69]. Said content can vary between 8% and 20%, depending primarily on the grain maturity [70,71]. The concentration and content of protein in grains impacts the specific rheological properties and durability of dough in the production of bread, cake, and pasta. Moreover, the production of good quality bread requires the protein content to be approx. 11.5% [72,73]. In our study, the protein content fluctuated between 11.99% (EMs, Bogatka) and 12.96% (Zyta, RLxEMs).

In another study conducted by Hucl and Chibbar [74], the concentration of protein in wheat grain collected form 33 cultivars was increased by between 12.6 (Canada Prairie Spring) and 17.4% (Hard Red Spring). The content of protein in triticale was within the range of 9.6% (Pernolec, nitrogen rate 90 kg N·ha$^{-1}$) to 12% (Humpolec, nitrogen rate 120 kg N·ha$^{-1}$) [75]. Under increased thermal stress, all the wheat genotypes showed an increase in protein content within the range of 12% (E-60) to 15.5% (E-65) [76].

The synthesis and accumulation of starch in grain directly impacts its yield and quality [77]. Between 60 and 75% of a wheat grain is composed of starch [78]. Depending on the grains' maturity, the starch content was within the range of 55.20 (early waxy phase) to 68.1% (full phase) [79]. In our own study, the starch content was within the range of 45.72 (Zyta) to 59.84% (Bogatka). In a study by Hucl and Chibbar [74], the starch content in the grains of the analyzed cultivars was between 64.6 and 74.2%. In another study on triticale grain, it was between 62.2 and 68.3% [75]. Under the conditions of thermal stress, an increase in starch content was observed in all wheat genotypes, varying between 62.9 (E-72) and 65.2% (E-60) [76]. Drought stress also impacted the content of starch (63–72%) and

gluten (8–12%) in wheat grains [80]. In our study, the gluten content fluctuated between 28.09 (Tonacja, RL) and 31.85% (Zyta, RLxEMs).

The exact mechanism of said effects has yet to be fully explained, as plant life continues to pose numerous mysteries. We know that light plays a special role in the process of photosynthesis, which affects the growth and development of plants. In our study, the use of red light and effective microorganisms, including, e.g., photosynthetic bacteria, as well as the combined application of the two factors, resulted in an increase in both the yield quantity and quality.

Notably, the methods analyzed in our study are environmentally friendly, which is particularly significant in the context of integrated agriculture.

To recapitulate, the literature reports pertaining to the impact of RL and EMs biopreparations on seeds point to a wide spectrum of factors. Consequently, rather diverse opinions can be encountered, emphasizing both positive and negative effects. Our knowledge on the subject remains patchy, despite the continued development of science and technology. The most common errors noticeable in research reports are mainly methodological, e.g., no indication of the type of soil, conducting experiments for only a single cultivar of a given plant, etc.

## 5. Conclusions

The conducted research showed that, in general, the use of RL and the application of EMs, as well as the combination of RL and EMs, positively influenced the tested values, although it depends on the variety. For the varieties Bogatka, Tonacja, and Zyta an increase in the values of gluten, SDS index, and germination was observed. In turn, in the case of starch, the Tonacja variety showed a decrease and the Zyta variety showed an increase. In the case of the grain yield for the cultivars Bogatka and Tonacja, an increase in the yield of grain and straw was observed, while for the cultivar Zyta, a slight decrease. The use of RL, Ems, and their combinations increased the protein content in all tested cases, except for the application of EMs for the Bogataka variety and RL for the Zyta variety.

The three-year treatment of soil with the use of EMs triggered changes in the soil characteristics. The content of humus decreased significantly in the EM-treated soil. Noticeable effects were also recorded in terms of mineral content. The presence of EMs reduced the soil content of phosphorus, potassium, and magnesium.

**Author Contributions:** Conceptualization, A.D.-H. and M.S.; methodology, M.S.; software, W.T.; validation, A.D.-H., M.S., and J.Z.; formal analysis, M.S.; investigation, A.D.-H. and M.S.; resources, S.M.; data curation, A.D.-H.; writing—original draft preparation, M.S.; writing—review and editing, A.D.-H. and M.S.; visualization, S.M.; supervision, J.Z.; project administration, S.M.; funding acquisition, W.T. All authors have read and agreed to the published version of the manuscript.

**Funding:** This research was funded by the University of Life Sciences in Lublin.

**Conflicts of Interest:** The authors declare no conflict of interest.

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
