# Peer review of "The Influence of Red Light (RL) and Effective Microorganism (EM) Application on Soil Properties, Yield, and Quality in Wheat Cultivation"

_agronomy, doi:10.3390/agronomy10081201_

Round 1

Reviewer 1 Report

In my overall opinion the model of experiment has not been planned properly. The Authors limited the  research to only one wheat cultivar – Bogatka.  I think that basing on an experiment with used single cultivar of wheat leads to unreliable conclusions.  Moreover, the Authors founded that the study on one cultivar is a the most common methodological error (L815-817). Additionally, the results of experiment, which are given as an average for three years, do not confirm that the observed trends were repeatable in each years of wheat cultivation. Besides the average values, the results for each cultivation year (average of three replications and standard deviation) should be given. The study should be extend to least three wheat cultivars.

The detailed comments are listed below:

L2-3: I suggest change the title of the manuscript. A significant part of the research concerns on the influence of effective microorganisms on soil properties - this should be included in the title of this work “

L29: Please specify which major nutrients are you talking about?

L84-87: Why the red light seed exposure was chosen? What may be the mechanism of the positive effect of this light on seeds? There is any information on this in the literature? Please extend the “Introduction” to those information. In this section too much attention has been given to the effects of EM but no information was provided about the influence  of red light.

L149: In the study the chemical composition of the grains did not analyze but the content of selected chemical components were determined.

L147-151: The impact assessment of the EM on soil properties was also the aim of the study –  this should be taken into account.

L154-155: An experiment was conducted on common winter wheat (Triticum aestivum, ssp. vulgare),cultivar Bogatka.  I think that the modification of the experiment’s model and  used only one cultivar is not a good idea. It is difficult to draft correct conclusions when we have results for one cultivar.

Please delete technology group. This classification of wheat is valid only in Poland.

L168-192: The device used for red light seed conditioning was described in details in publication:  “Impact of Pre-Sowing Red Light Treatment of Sweet Corn Seeds on the Quality and Quantity of Yield” – please provide overall description and referee to this publication. 

L194-196 – Which means  EM-A – please explain this abbreviation. The seeds were soak immediately before sowing? - please clarify.

L204-205: What methods were used to determine the bioavailable forms of P2O5, K2O, Mg and bioavailable forms of microelements?

L205: Please give producer of the  spectrometer.

L318 :  change on “…. 100 g of soil by 0.1 mg…”,

L319 -320: “unlike” ? - rather “similar to phosphorus”

L320 and L322: Please give correct value in  %

L322: Change the comma in  “0,65 mg”  on a dot.

L427 Table 5: The lowest value of  gluten was noted for 1st object – please designate with letter “d”

L754 -755: Why the starch content is so low? – please explain obtained results.

L817: The authors indicate that the most frequent methodological error is “conducting experiments for only a single cultivar of a given plant”   and in this study  they made this mistake.

L 820-832 The conclusions should be redrafted taking into account of  the results at least three wheat cultivars.

L831-832- This conclusion does not provide any information.

Reviewer 2 Report

Line 19-20: you stated that "The aim of the study was to determine the impact of red light (RL) and effective microorganisms (EM) on wheat". Please define on what wheat properties. Growth, productivity, physical-chemical...? Where did you apply this, in the field or in silos...?

Line 155: Please insert the reference for the claim that it belongs to technology group B. 

In m&m please explain were the samples treated with both procedures (ER and RD) or were they divided into two groups and one was treated with ER, one with RL + was control. 

Round 2

Reviewer 1 Report

Dear Authors,

the detailed comments are given in the text of the manuscript (attached file). 

Author Response

This manuscript is a resubmission of an earlier submission. The following is a list of the peer review reports and author responses from that submission.

Round 1

Reviewer 1 Report

Authors evaluated effects of light type and microorganisams preparation of wheat grain.

Title: it should be put first full phrase – efect. microrganisms  and then EM in parantheses

Keywords: do not use sam words as in title

ABSTRACT: Line 16: three not tree, there are lot of these type of mistakes in article

Simplified the language in abstract and in all manuscript.

INTRODUCTION:

Line 50: there are not any info what all studies you referenced here found regarding seeds,

Line 54: paragraph about EM should be rearrange, it seems part like introduction and part M&M

Line 71: ...improvement specifically in the Polish context…. This is not fair written and used English, manuscript should be checked by native speaker

MATERIALS: Line 110: Sentence has reference and it is nothing about Figure 1, change.

Line 120: Why you started here with sowing, make understandable sections: plant material + cultivation, light nad EM treatments, soil analysis (it is not only texture analysis), grain quality analysis…

Line 123-126: It is not important where you did analysis but what methods you used.

Line129: You wrote about culktivation and than about germination in Petri dishes, male this logical.

RESULTS: Silt is clay, dust is silt

Line 165: what means: The content of phosphorus in the soil was found to have been average?

Table 4. why you have < before Cd, is that under detection limit?

Table 5. too complicated, if we have all numbers, do we need to have relative values, what about significance?

Line 204. pre- swing?, many this kind of mistakes

Line 206: this paragraph should be rearrange in simple way: eg.  Cultivar Bogatka has positive reaction on EM, while Tonac. And Zyta had yield reduction with EM treatment.

Do not use object, it is treatment.

Line 275: said content?

Discusion: Line 305: Dead soil? Upset micro balance?

Why you do not use EM, when you stareted with acronym use it until end.

Too decriptive, did you give any explanation what possible mechanism should be responsible for your findings.

Reviewer 2 Report

Major remarks:

The title used for this study is not really explicit. This title must be improved in order to give the reader a precise idea of the objective of the study presented.

The bibliographic part is still too general; information concerning the hypotheses on the mechanisms implemented during this phenomenon of biopreparation are lacking.

According to the authors, “the aim of the study was to determine the impact of pre-sowing seed conditioning with the use of red light and an EM biopreparation on the germination rate, grain and straw yield, and chemical composition of wheat grain” (lines 84-87).

However, on reading the results part, it seems that, in this specific study, the impacts of pre-sowing seed conditioning techniques [i.e. red light and/or Effective Microorganisms (EM)] are combined with a soil pretreatment (or not) by an EM biopreparation? This is the reason why the authors must distinguish more clearly between the supposed effects of soil conditioning (EM vs. 0) and pre-sowing seed conditioning (N, EM, N x EM vs. C) on the different parameters selected here (yield, grain composition and quality).

It is therefore essential to rework the results part in order to clarify the explanations given: (i) general effects of soil conditioning (ME vs. 0), (ii) general effects of pre-sowing seed conditioning (N, ME, N x ME vs. C) and finally, (iii) the possible genotypic effects. In this context, it would be interesting to propose a new presentation (clear and synthetic) of the results in Table 5 (and Table 7), because the reader has great difficulty in extracting essential information from them.

In the paragraph 3.4, how authors can explain the potential role of a pre-sowing seed conditioning on the germination capacity of wheat grains produced by mature plants?

In wheat grains, protein and starch synthesis/accumulation are very characteristics. In fact, from anthesis, nitrogen necessary for the synthesis and accumulation of grain proteins comes mainly from remobilization of sources of leaf nitrogen (i.e. proteolysis of leaf proteins) while, at the same time, saccharides necessary for the synthesis and accumulation of starch in the grain come almost exclusively from post-anthesis photosynthesis (i.e. de novo synthesis). These two metabolisms are therefore opposed by nature. It would therefore be interesting that authors take these elements into account when commenting on the results obtained in Table 7 (page 9).

Finally, at this level, it is important to remember that relationships that can exist between the biochemical composition of wheat grains and their aptitudes for technological transformation are not simply quantitative relationships! Thus, since the 1990s, the scientific community has been able to demonstrate the existence of significant qualitative relationships between the structure of prolamins (Gliadins/Glutenins, HMW-GS/LMW-GS, Molecular Weight Distribution of Glutenins…) and/or accumulated starches (Amylose/Amylopectin, Starch Granule Distribution…) and the wheat grains aptitudes for processing. This is generally the reason why this qualitative component is taken into account by integrating several genotypes during the experiments.

The formation of a gluten network and / or the solubility of proteins in SDS are therefore not linked to strictly quantitative effects. Thank you to the authors to take this into account when presenting their results and conclusions.

The discussion part of this study must be really enriched with elements which allow the authors to formulate mechanistic hypotheses (lines 335-342). In addition, this discussion sets out a number of conclusions which do not really conform to the results presented. As an example, here is a sentence to modify:

Lines 315-316 : “In our own study, the combined uses of effective microorganisms and light treatment resulted in 46% (Bogatka) and 58% (Tonacja) increase in grain yield, and a 63% (Bogatka) and 83% (Tonacja) increase in straw yield.” No, all these results are obtained for wheat samples corresponding to a EM pre-sowing seed conditioning when seeds are sown exclusively in field treated with EM after 3 years (Table 5). There is no combination with any light treatment here!!

Finally, it would seem that confusion is made at lines 350 and following. Indeed, the authors seem to assimilate the effect of a direct light treatment of a seed on its germination with the treatment they applied to a seed before sowing and the germination capacity of the wheat grain harvested several months later. ? I think these assumptions need to be completely revised.

Other remarks:

In the "Plant material" section, the authors state that three different wheat genotypes have been selected for this study (line 94). The reader would like to know the reasons which presided over this choice? Do these genotypes differ from their agronomic behavior, from their technological behavior...?

In the paragraph 2.2 (lines 98-101), the EM pre-sowing seed conditioning technique is not defined and in this section the EM solution used in not defined/referenced (line 98 and 116). Finally, Figure 1 (line 112) does not really give an idea of the light device used here. Thank you in advance to the authors for making an educational effort.

In the "Yield and grain quality" paragraph (line 129), the authors are very vague about the cultivation conditions applied during these experiments. The reader would however like to have more details on these.

In the paragraph "Chemical composition of wheat grains" (lines 135-138), it would be interesting to explain why among the different criteria used to assess the protein quality, authors selected the index of sedimentation and the gluten %?

Although it is the same experimental design, the authors chose a different presentation for table 5 and table 7. For what reasons did the authors replace the columns "O" and "EM" (table 5) by the columns "After the 1st year" and "After the 3rd year" in table 7?